# A Comprehensive Survey about Thermal Comfort under the IoT Paradigm: Is Crowdsensing the New Horizon?

**DOI:** 10.3390/s20164647

**Published:** 2020-08-18

**Authors:** Valentina Tomat, Alfonso P. Ramallo-González, Antonio F. Skarmeta Gómez

**Affiliations:** Faculty of Computer Science, Universidad de Murcia, Campus Universitario, 30100 Murcia, Spain; alfonsop.ramallo@um.es (A.P.R.-G.); skarmeta@um.es (A.F.S.G.)

**Keywords:** IoT, Crowdsensing, thermal comfort

## Abstract

This paper presents a review of technologies under the paradigm 4.0 applied to the study of the thermal comfort and, implicitly, energy efficiency. The research is based on the analysis of the Internet of Things (IoT) literature, presenting a comparison among several approaches adopted. The central objective of the research is to outline the path that has been taken throughout the last decade towards a people-centric approach, discussing how users switched from being passive receivers of IoT services to being an active part of it. Basing on existing studies, authors performed what was a necessary and unprecedented grouping of the IoT applications to the thermal comfort into three categories: the thermal comfort studies with IoT hardware, in which the approach focuses on physical devices, the mimicking of IoT sensors and comfort using Building Simulation Models, based on the dynamic modelling of the thermal comfort through IoT systems, and Crowdsensing, a new concept in which people can express their sensation proactively using IoT devices. Analysing the trends of the three categories, the results showed that Crowdsensing has a promising future in the investigation through the IoT, although some technical steps forward are needed to achieve a satisfactory application to the thermal comfort matter.

## 1. Introduction

The Internet of Things (IoT) is one of the greatest revolutions that have emerged in the last decades. IoT is based on the concept of *everything connected everywhere*, which is transforming the modern world [1]. This new paradigm aims at converting real-world objects into smart objects [2], i.e., objects that communicate via the internet to create a common infrastructure that connects human-to-human, things-to-things and human-to-things [3]. IoT has rapidly started to involve multiple fields, such as healthcare, industry, urbanism, home appliances and so forth [4], reaching not just electronic devices but also merchandising, furniture, culture, landmarks and even food and clothing [5].

In the fight against climate change and energy waste, buildings are often mentioned as one of the primary sources of emissions due to their large energy consumption. Studies confirm that 20% to 40% of energy consumption in developed countries is attributable to the building sector, and more specifically, 50% of building consumption is due to Heating Refrigeration and Air Conditioning (HVAC) energy use [6,7]. When looking at the use of HVACs, people’s thermal satisfaction plays a key role, because of the importance to find a balance between energy use and users’ thermal comfort [8].

Research groups have developed several procedures to apply IoT to the issue of thermal comfort, with points of view that differ greatly from each other. Consequently, here, the authors propose a review of the most influential scientific publications that mix the fields of the IoT and thermal comfort, so the authors create added value to the scientific community that will identify the streams that have been generated naturally. In this review, the authors identified three main ways in which the IoT can be applied to thermal comfort studies: through hardware, through simulation and, still marginally, through the new Crowdsensing paradigm.

This literature review focuses in particular on the role played by the occupants, describing how researchers are moving toward a people-centric approach in which the user’s opinions and preferences matter. This point of view is slowly streaking ahead as opposed to the “one-fits-all” perspective, in which people were passive receivers of a thermal environment chosen by standards.

### 1.1. Context

According to The National Human Activity Pattern Survey, conducted in 2001 but still widely considered among the scientific community, in modern society, people tend to spend 87% of their time indoors [9]. Identifying an indoor thermal zone in which people are comfortable is an ever-current topic. The modern perception of the problem started in the 1970s, presenting two different schools of thought from the beginning. On the one hand, an initiator of the field of thermal comfort studies was Fanger, who introduced a steady-state model to relate thermal sensation to subjective parameters, such as activity and clothing levels [10]. Since the introduction of Fanger’s comfort model, who obtained his results through experimentation in climate chambers, the accepted parameters to evaluate thermal comfort have been: indoor air temperature, mean radiant temperature, indoor relative humidity, internal air velocity, clothing insulation and metabolic rate. These parameters are incorporated in the so-called Predicted Mean Vote (PMV), an index which aims to evaluate and predict the state of thermal well-being of occupants. This method is based on the heat balance of the human body that, as explained by Höppe [11], comprehends both the physical heat transfer process and some thermophysiological mechanisms such as sweating, shivering, etc. Fanger’s model introduced the aforementioned parameter called Predicted Mean Vote (PMV) to also relate thermal sensation to subjective parameters, and a parameter called Predicted Percentage of Dissatisfied (PPD), to estimate how many occupants would feel thermally dissatisfied. Subsequent studies showed that the PMV model fails to describe comfort temperature outside of the exact conditions of Fanger’s experiment, because it does not consider the effect of people’s adaptation. For this reason, Fanger and Toftum [12] elaborated another model which include an expectancy factor, *e*. On the other hand, Nicol and Humphreys [13] supported that experimentations in the climate chamber do not consider other complex parameters, such as time sequence and social factors, that are fundamental in the study of thermal comfort. It was the beginning of the so-called Adaptive Thermal Comfort Model. Supporters of the latter have also been de Dear and Brager [14], who stated that in naturally ventilated buildings, the range of accepted temperature depends in great part to other parameters, such as external temperature and people’s adaptability. Also, according to more recent studies, in the case of free-running buildings and in mixed mode ventilated buildings, i.e., in case people can adjust or change their environment with operable windows or through metabolic alterations, researchers tend to consider the use of the steady-state model inadequate [15,16]. Besides, the adaptive comfort standard provides thermal design guidance and indicates the acceptable and the optimum indoor temperature ranges depending on the climate zone [17,18].

Today, both models are regulated by ISO 7730:2005 [19], EN 16798-1:2019 [20] and ASHRAE 55:2017 [21] regulations and are accepted and used worldwide, although significant differences among them keep being underlined [22]. It is worth mentioning that current experiments are investigating a novel model, that aims to explain the effect of short-term temperature changes on human’s thermal comfort, called alliesthesia. According to this research line, thermal comfort is conceived as dynamic, and the environment as transient and non-uniform [23,24,25]. Recent studies included other parameters in the perception of thermal comfort, i.e., thermal alliesthesia and habituation, so that new models are starting to include both a static and a transient component [26].

In this context, IoT represents a great opportunity to face the issue of thermal comfort in buildings. In the last decades, IoT-based systems have already become a fundamental ally in proposing new smart solutions to ameliorate energy consumption in buildings [27]. Taking a step forward, IoT applied to thermal comfort allows to consider the energy-matter not only passively from the point of view of the building itself, but also in its relationship with the user. Although this review does not aim to investigate this energetic topic, in many studies, thermal comfort is presented in relation to energy efficiency, hence its incorporation in this document.

Furthermore, several studies demonstrate that thermal discomfort has effects both on health and human performance. Lan et al. [28] conducted some neuro-behavioural tests on users doing office work activity to monitor their productivity. It was possible to measure a quantitative relationship between productivity losses and thermal sensation and to conclude that thermal discomfort leads to reduced performance, both in case of feeling too cold and too warm. In another study, Lan et al. [29] showed that in thermal discomfort, users reported the so-called sick building syndrome (SBS) symptoms, as well as a negative influence on the mood, the willingness to make efforts and the air quality perception. It is noteworthy that the negative effects on health and performance are not related to the distraction of subjects, but to physiological reactions to discomfort, i.e., they will occur even if subjects have become adaptively habituated to an unfavourable environment.

IoT technologies bring a technological ecosystem that has to be understood to be aware of the role of these systems in the buildings. These principles more based in computer science tend to follow a common structure. Authors have used this common structure to help to classify the different works found in the literature. IoT in buildings is composed of three layers [30]:Sensing or Perception Layer, dedicated to the acquisition of information.Network Layer, that connect data and manage the control centre.Application Layer, that is supposed to achieve the energy management.

From the literature, it is possible to see that the studies on IoT and thermal comfort differ from each other because of one or more of these layers but maintain the general structure. For what concerns the first layer, the sensing process through the use of IoT can collect data through sensors both from the environment (in fixed locations) or from users (with personal devices carried by the participants). The literature considered also differ in the way of analysing data, although the most recurrent methods are mathematical tools and virtual modelling. About the networking, in the literature available, its complexity can vary from basic sensor networks to complete architectures that incorporate different components.

The application layer consists in the generation of outputs that, like in the sensor layer, could involve the environment, i.e., a change in setting the air temperature, or the single persons, with personal devices suitable for improving the thermal comfort of each one.

These differences also reflect the semantic duality in the conception of the IoT itself. Visions of the IoT often embraced a “Things oriented” perspective or an “Internet oriented” perspective, intending to create a bridge between the physical and digital world [31]. Taking in mind this duality, it is possible to identify studies that involve physical smart objects among thermal comfort literature, like smart HVAC systems, personal devices and studies based on virtual modelling and simulation. In this review, the two categories will be referred to as “studies with IoT hardware” and the “Building Simulation Model”. Of course, one method does not necessarily exclude the other. So, besides these two approaches, the authors will present a few studies that use data to validate the computer model simulation, and others that propose a unified platform to combine the methods.

However, in both cases, often people are considered as passive parts of the system, one variable of the study out of many others. Differences among individuals are not considered enough and people’s response to different stimuli are often inaccurate or undermined. For these reasons, in the last decade, studies have developed a new approach to the thermal comfort matter, in which the main focus is people’s opinion and perception: *The Crowdsensing paradigm.*

“At present, public groups are frequently portrayed as ignorant or irrational in the face of scientific progress” [32]. In the scientific dimension, on most occasions, “the crowd” is seen as a passive receiver of progress’ benefits or established standards. Often, users are tacitly seen by scientists as ignorant, misled or simply contrary. Even so, according to sociologists such as Williams [33], it is important to form a citizen’s view of science, i.e., to create a sustainable development that foments a people-oriented perspective. This is particularly relevant in the investigation about the energy behaviour of buildings, in which the final aim is users’ thermal comfort. Among people’s thermal preferences, there are differences both between two people in the same environment on the same occasion (inter-individual differences) and between the same individual in the same environment on different occasions (intra-individuals differences) [34]. That is why the extension to the thermal comfort of the new paradigm of Crowdsensing, based on every individual’s perceptions and impressions, is utterly urgent. As Erickson and Cerpa [35] stated: “At the individual level, occupants are 100% accurate when determining if they are comfortable—only their opinion matters”.

According to Ganti et al. [36], devices that are customer-centric, like smartphones and music players, will be the evolution of the IoT, allowing an upgrade to a societal scale.

Crowdsensing could be classified, depending on users’ involvement, into (1) participatory and (2) opportunistic sensing. In participatory sensing, people are directly involved, through reporting data, taking pictures, etc. In opportunistic sensing, the involvement is minimal, the explicit action of users is not necessary. A clear example is user-generated data in mobile social network services [37], that helps to understand the dynamics of our society.

Before the concept of Crowdsensing appeared, Dutta et al. [38] explained the problem and the necessity of this kind of solution well: “Mobile participatory sensing uses consumer electronics to capture, process and disseminate sensor data, complementing alternative architectures by ‘filling the gaps’ where people go but sensor infrastructure has not yet been installed”. It is clear that thermal sensors cannot be everywhere. In public places and workspaces, it surely is more common, but in most of the locations that users visit, the “gap” is unavoidable. In other words, as a way to extend researchers’ results to people, one should consider other parameters to measure the thermal comfort. Nicol and Humphreys [39] explained this well: “And finally… do we really need to specify indoor climate? Given a full understanding of the mechanism at work, it may eventually be possible to produce thermal standards for building which do not resort to specification of the indoor climate. The characteristic of a building in relation to the local climate may be sufficient.”

One of the most fundamental constituents of the IoT is domotics, or home automation, popularly called smart home. Domotics refers to several fields, namely smart appliances, home entertainment, control and connectivity, comfort and lighting, security and, of course, energy management. Smart home, that has to be intended broadly and not just referred to in a residential context, is an incredibly fruitful market sector, considering that just in the last three years, it has doubled its sales and they are expected to grow by 60% by 2024 [40]. Up to now, almost 95 million homes are considered smart from an energetic point of view, thanks to the diffusion of new technologies affordable for the majority of people (or at least in wealthy countries).

Smart home is a trend from 2013, but it reached its higher diffusion in the last years, due to the commercial success of devices such as *Amazon Alexa*. There is something revolutionary in it, since in a very short time, IoT switched its status from news of scientific interest to fashionable for users. Alexa is an intelligent virtual assistant that works through a smart speaker, *Echo*. Behind the basic functionality of the speaker itself (plays music, reads audiobooks, streams podcasts etc.), what is interesting in its configuration is the capability of controlling other smart devices. In other words, Alexa can be considered itself a home automation system, which interacts with people while wiring all the smart objects around it. It was released in November 2014, but it started to be universally popular from 2017 [41], probably due to massive advertising campaigns that gave a sort of sentimental feeling connected to its use, such as improving family members’ relationships, summarized by the concept of “sharing is caring” [42]. In January 2019, Amazon stated that they sold an impressive amount of 100 million Alexa devices [43]. Simply, it has been a revolution that shifted people’s attention to domotics. Figure 1 is a graph created by the authors using Google Trends data [41] that shows how much Alexa influenced the world of home automation: peaks in people’s interest almost perfectly overlap since the end of 2016.

In Figure 1, the comparison also includes the trend of Google Nest. It does not affect the diffusion of smart automation as much as the commercial phenomenon of Alexa, but still, it helps to explain some peaks of penetration. In fact, Nest is another trend in the modern market, being the most popular smart thermostat worldwide. Nest is not just a programmable thermostat—it learns from every user’s preferences, being able to program itself as a consequence. It also learns about the time needed for a specific home to warm up or cool down, managing a proper schedule for every situation. Besides, Nest has a raise awareness aim, inasmuch it shows users when they are in an energy-efficient mode [44]. It can be said that Nest learns from us and we learn from Nest. Giving a brief context, Nest was born as an invention of Tony Fadell and Mark Rogers who, in 2010, founded the Nest Labs. The brand immediately had a discrete success, and Google bought it in 2014 and it became even more popular, helped by other factors such as improvement in wireless broadband technology’s diffusion and the first standards about smart meters’ requirements in offices and homes.

Several other brands deal with the energy topic in home and office automation. Up to now, the main competitor has been Honeywell, connected with Apple Homekit (the famous voice assistant *Siri*), but recently, also Ecobee, that has a built-in Alexa Voice Service and room sensors capable of detecting occupants [45]. Other famous enterprises like Siemens, Samsung, Xiaomi, KeenHome, etc., are trying to take a piece of this market sector but, as shown in Figure 2, the magnitude of Nest and Honeywell is hard to reach.

It is worth noting that there exists a gap between types of applications in the market and the effective use of people, i.e., something is hindering the utilization of IoT in homes and workplaces. Although they will not be explored in this paper, researchers identified the potential reasons, namely energy consumption, device connectivity, safety and, of course, security [46]. The most common concern is cybersecurity: people are afraid of being spied on in their own homes and workplaces and this is a big obstacle in the diffusion of the aforementioned devices. Users agree about the devices’ function of making their lives simpler, but they do not want to introduce new risks [47].

### 1.2. Paper’s Structure

The thermal comfort studies in the last decades have been strictly related to the IoT development. Every technological advance in the IoT world corresponded to a big step forward in the understanding of the comfort conditions. Growing accuracy of the sensing tools, new technological elements in the loop and more powerful simulators marked the development of modern thermal comfort investigation. The central question of this work is understanding whether or not the novel paradigm of the IoT, the Crowdsensing, can represent a future path in the investigation about thermal comfort. Studies based on thermal surveys are normally conducted in situ, mainly because of the necessity to measure environmental parameters with proper devices. Crowdsensing broke this physical limit for other fields of study, so it would be an enormous improvement to extend its potentialities also for thermal comfort, allowing the researchers to conduct their surveys remotely.

To answer this question, the method that seemed more appropriate was analysing both how the IoT was applied to the thermal comfort study in the last ten years and how the Crowdsensing is being applied to the other fields of study.

The research methods of the articles for this review follow this duality. Applications of IoT to buildings and to energy efficiency have been the focus of the authors’ research group for years, so the starting points were the articles that the authors considered inspiring or noteworthy. It was a natural consequence noting that many of these articles mentioned the same researchers or the same investigation groups, and the main contributions were found. Then, research through academic platforms, mainly Scopus and partly Google Scholar, completed and confirmed the results.

The approach for the Crowdsensing part was quite different. As the phenomenon is quite new and not universally known, the starting point was searching for articles about Crowdsensing in general, through the mentioned academic platforms. The most cited papers were selected and the major contributions in terms of explication of how the paradigm works. From this base, articles about application to the thermal comfort topic were searched, though the keywords “Crowdsensing + Thermal Comfort”, “Crowdsensing + IoT + Thermal Comfort”, and so forth. The research was unsuccessful: the Crowdsensing application to thermal comfort was not investigated previously. After trying with “Crowdsourcing + Thermal Comfort” and “Participatory sensing + Thermal Comfort”, some results were obtained: something was moving towards that direction.

The paper is structured as follows. Section 2 investigates the thermal comfort studies with IoT hardware. The section presents a first part in which studies that propose an improvement in thermal comfort conditions are analysed, both obtained through changes in the whole environment and in the micro-environments around occupants. The second part of the section presents articles that use hardware with monitoring aims, as demonstration studies or sustained by experimentations.

Section 3 explores the mimicking of IoT sensors and comfort using Building Simulation Models. The first part of the chapter simulates variations in thermal comforts due to changes both in the physical surrounding and in users’ behaviour. The second part of the section proposes studies which incorporate the simulation tools in a more complex IoT architecture, and collects studies that simulate thermal comfort in future climates. Section 4 describes the new Crowdsensing trend and explores its application to other fields of study. Tools to involve people in the loop are briefly explained and the rest of the chapter is dedicated to the first applications to the thermal comfort matter. Section 5 shows quantitative results, mainly in terms of analysis of the scientific community’s interest in these subjects. This is useful to understand if the Crowdsensing paradigm has space to grow as a scientific topic and as an application to the thermal comfort. Section 6 contains discussions about the main topics. Section 7 presents possible directions for future works.

## 2. Thermal Comfort Studies with IoT Hardware

In this section, the authors analyse how smart objects are used to study thermal comfort and energy efficiency. Smart objects are considered part of the IoT when they are interconnected among them or to the internet. Hence, within the object, there is a necessary embedded system that acts as a connector. The literature shows several approaches and solutions that have this kind of IoT application in common.

After performing a comprehensive analysis of the literature available, it has been chosen to group studies into solution strategies. As explained in previous sections, the mass of the publications verse on how to improve the energy efficiency and to maximise the thermal comfort of individuals. Among the literature, the authors identified studies which aim at transforming the surrounding environment and studies which work on directly modifying the thermal sensation of people through individual devices. Besides, there are studies which do not present a solution, being centred on monitoring and analysing the phenomenon. Among them, the authors distinguish studies which are validated by experimentation or simulation, and studies that focus on conceptualising the problem and design an IoT architecture. Table 1 and Table 2 summarise the contributions below, and the following four sub-sections show and describe in detail the works on this topic.

### 2.1. Strategies Involving IoT Technologies to Produce Changes in the Indoor Environment

It is quite common in the literature to give the users the control over the HVAC system to change the thermal environment according to the preferences of the majority. Van Hoof [48], who analysed 40 years of thermal comfort literature since the introduction of Fanger’s PMV, concluded that thermal comfort for all can only be achieved with the effective control of occupants over their thermal environment. This is a democratic solution primarily in open offices, where several workers have to share the same space.

Feldmeier and Paradiso [49] proposed a personalized HVAC system with four components: a central network hub, portables nodes, control nodes and room nodes. The users’ effort required is minimum, as they only have to press a button on their wrist-worn sensor node if they feel uncomfortable, while physiological data are also collected. The project reached an 80% thermal comfort goal, attributing discomfort to a bad distribution of temperature in rooms and achieving 24% of energy savings.

Sung et al. [50] presented a complete IoT architecture, composed of several components for each layer (Perception Layer, Network Layer and Application Layer, as discussed in the Introduction Section), including sensors and a smart terminal for every device. The process involves the use of Visual C# software for the monitoring interface, and Matlab for analysis and simulation. The study proposes three control modes that affect the set-point of smart HVAC devices, which include air conditioning, electric fan and humidifier, to obtain changes in the environmental conditions. The three control modes presented are Comfort mode, General mode and Energy-saving mode, that differ in the values of ambient temperature, relative humidity, indoor wind speed, expected PMV (from calculations), expected PPD (ditto) and estimated energy-saving. According to the authors, the expected target PMV range, that is set equal to 0, 0.5 or 0.7 depending on the control mode, can be achieved in all three modes. The same authors resumed the architecture to conduct a similar study [51] in which a questionnaire about satisfaction is included. In this case, the main evaluation parameters were the maximum variation and the average temperature difference between users and the environment. These values are affected, for instance, by movements or proximity to the door. In this study, the three cases considered are uncontrolled, constant control and fuzzy control. They create a thermal environment according to the simulations and they ask for the opinion of respondents, obtaining 10 out of 12 people satisfied.

About smart set-point in HVAC, it is also worth noting a study of Ramallo-González et al. [52], which will be presented in Section 4.2 since it approaches both topics.

### 2.2. IoT Devices to Modify the Micro-Climate of Each Individual and Enhance Thermal Comfort

The strategies presented in this section start from the assumption that the standard deviation of preferred temperature in individuals at rest can differ up to 2.6 °C [53]. If one also considers individual differences in clothing choices and metabolic activity, the gap can be even greater. This is why the authors of the works that are described in this subsection stated that it is unlikely that with a single thermostat, thermal comfort can be reached by everyone, and that personal devices are necessary.

A study of Knecht et al. [54] considered the use of personal wearable devices to investigate thermal comfort in workspaces and to suggest a possible solution to discomfort. Garments both for heating (gloves, shoulder pad, socks, etc.) and cooling (for wrists, ankle, a body wrap, a necktie, etc.) are connected to a radio circuit to transmit device temperature information, which is a proxy of the user’s thermal status. On the one hand, participants presented some complaints mainly about the aesthetics and comfort of wear. For instance, for cooling garments, a sensation of discomfort was reported since people’s skin is more sensitive to direct contact with a cold source than the opposite. On the other hand, users stated to prefer personal devices to avoid the implication of their thermal preference on the others, in the case of a shared thermal environment: adjustments on a local level were preferred compared to global scale ones.

Another example of personal equipment, although not wearable, is given by Zhang et al. [55] through a study based on the use of foot warmers connected to the internet (smart). The research was conducted in a workspace of Berkeley University and it demonstrated that using personal foot-warmers while lowering the HVAC system’s set-point did not affect the occupants’ thermal comfort. The savings were up to 75%, depending on the outdoor temperature. The most impressive result of the experiment was the finding of a satisfaction rate of 100%, a unique goal in comfort field studies, but they only used the system for heating.

One of the most noticeable contributions is given by Kim et al. [56]. A large number of studies start by comparing the traditional thermal comfort models to choose the most appropriate. Instead, Kim et al. presented a new critique of the existent models and their limitations, proposing their personal solution to overcome their shortcomings. They pointed out that both PMV and adaptive models have poor predictive accuracy when applied to small groups of people and that they are based on input variables that are difficult to measure in the real-world, being necessary in most cases simplifying or assuming them. Moreover, it is not possible to aggregate other variables, such as body mass index, and they cannot reflect conditions in a specific setting, different from the original dataset. This critique summarizes well the objective reasons why a new people-centric approach is necessary. The Personal Comfort System (PCS) proposed by Kim et al. consists on using smart chairs developed at Berkeley with embedded sensors, connected wirelessly with a controller, that are able to both heat and cool. The chairs have been used by 38 occupants over seven months. They compare six machine learning algorithms to analyse the users’ thermal preferences. This system obtained a 69% gain in predictive accuracy compared to traditional comfort models so it can represent a potential replacement in feedback surveys. This opens a new avenue of research and could be the third big family of thermal comfort theory, after Fanger’s and Humphreys’ theories.

### 2.3. Studies with Monitoring Purpose Using IoT Technologies

In some cases, studies are aimed at analysing a situation in terms of thermal satisfaction and indoor thermal conditions, but without presenting a proposal for improvements. Often, they also conduct experimentations to test their system, simply omitting the Application Layer. These studies are not be presented in Table 1, as they have no material results in terms of people’s satisfaction nor energy efficiency, but they are summarized in Table 2, in which accuracy of the methods is compared.

An example is a study of Salamone et al. [57], which based the Sensing Layer on the use of both wearable and the so-called *nearable* devices, a word coined to indicate sensors that refer to a micro-environment, like a single workspace. They chose wristbands as wearable devices and low-cost sensors located at 40 cm from the participants as nearable devices. Data are processed using a noise-detection algorithm which detects the electro-dermal activity of respondents. Besides, the evaluation of thermal satisfaction is done through a web platform survey. Parameterising some parameters with the software Grasshopper and processing data through a Python script for Machine Learning, the authors managed to define the optimal personal thermal comfort with good accuracy. Then, the aforementioned authors tested their integrated method in another study [58], presenting results based on the PMV. Although there is no proposal for energy-saving or improvement in users’ thermal satisfaction, these studies certainly have their merit on considering an environment that describes the individuals’ micro-environment. In fact, often, room sensors are located in spots that do not correctly represent the thermal condition of individuals. In some cases, as in co-working spaces, it is just insufficient for a unique sensor to report the thermal surroundings of every workstation and, in other cases, sensors have a wrong position. So, the solution, which combines wearable and nearable devices, could be seen as ideal to investigate the specific environment of each user. The work shows that the comfort zone of every user can vary considerably both in terms of operative temperature and relative humidity.

Many research works consider that collecting users’ feedback is unpractical in the long term, so they propose an approach based exclusively on physiological data. Ghahramani et al. [59] proposed a method to study thermal comfort based on infrared thermography of the human face. In fact, the human body also responds to thermal stress through the cutaneous vessels, that are particularly dense around the human face. Monitoring the facial skin temperature as a monitor of skin blood flow, it was possible to analyse the different reactions to thermal stimuli. To do so, an infrared sensing system was installed on a tan eyeglass frame, collecting data from four different parts of the face (namely ear, nose, front face and cheekbone). Then, they developed a hidden Markov model (HMM)-based learning method that can be used in future studies. The validation of the method was conducted with the feedback of ten test subjects, for a total of 457 votes, of which 87 were uncomfortable and 370 were comfortable. The average accuracy for predicting uncomfortable votes resulted to be 82.8%.

Also based on physiological data collection is a study of Dai et al. [60], who developed a predicting model based on skin temperature. Data were collected through Fibre Bragg grating (FBG)-based sensors positioned on 13 body locations. They conducted two experiments, a group one with 11 subjects in a Controlled Environmental Chamber at UC Berkeley and an individual one on a male occupant in a private office in China, to validate the method. Besides the physiological data collection, the whole-body thermal sensation was investigated by pop-up questionnaires (via computer). The sensation scale used was a 9-point scale, in which the points called “very hot” and “very cold” were added to the traditional ASHRAE 7-point scale, and the Thermal Sensation Vote was simplified into 3 groups (“heating demand”, “neutral”, “cooling demand”). The results showed an 80% accuracy with a single skin temperature, if the clothing level was controlled. For the model with two inputs, they suggested to put the sensors on the shin and upper arm. They proposed an intelligent control method based on a support vector machine, concluding that the linear kernel was preferred to the Gaussian Kernel. Moreover, predictions resulted to be more accurate for heating demands than for cooling demands.

### 2.4. Demonstration Studies

A common approach in IoT-for-comfort research consists of designing the IoT architecture behind the idea of an IoT environment for comfort, but in which experimentation, and in some cases simulation, are not contemplated. The studies’ perspectives keep being interesting and some ideas are well-structured and/or have great potential, so it is worth mentioning them.

Park and Rhee [61] created a complete smart building system, taking care of the three layers of the IoT system. The equipment includes electrical devices and application, smart HVAC systems and personal wristband and smartphone application. This is achieved through two different models in Matlab/Simulink: the first one is a static model that controls the heating system, and the second one is a dynamic model which achieves the control for users’ thermal comfort. The dynamic model created is based on the heat balance equation of the human body, in which heaters will be controlled by the PMV value. In particular, the PMV is requested to range from −0.2 to 0.2, and when it reaches these extreme values, a cooling or heating system will be activated.

Some interesting contributions are given by demo abstracts, i.e., papers that just express an idea, without developing it.

For instance, Krioukov and Culler [62] presented an experimental prototype of personal building control based on three components: a smartphone application, building controls and building sensors. The feedback through the app directly corresponds to a change in a smart HVAC operation. Although quantitative results were not presented, the authors stated that the achievement was expected to be a larger range of thermal comfort for the occupants. Another demo that is worth mentioning is the study by Ploennigs et al. [63] that combined the concepts of thermal comfort and virtual reality, allowing the users to visualise in real-time the environmental parameters collected by the sensors, for the first time making the participants fully aware of the conditions they are in. This method created an immersive understanding and interaction of users with the systems that affect their thermal perception.

Likewise, Choi et al. [64] conducted a series of experiments on 18 subjects to investigate the correlation between the whole-body thermal sensation and the local body skin temperature. A data acquisition (DAQ) system was designed and used to collect environmental data and user preferences, while an exposed thermistor-type sensor was used in order to measure skin temperature in 8 different body segments. For the thermal sensation expression, the 7-point scale of ASHRAE standard was used. A stepwise regression analysis was performed to choose predictive variables by automatic procedures. The study revealed that the whole-body thermal sensation is significantly correlated to the local body skin temperature, with differences due to gender and body mass index (BMI). The arm and the wrist resulted to be the segments with a stronger impact. According to the authors, the maximum accuracy would be obtained using data from that waist, arm and wrist, together with information about gender and BMI. The estimated overall sensation was 95.87% accurate. A practical alternative is proposed, which associates a changing rate of skin temperature to a body point or segment. In the case of considering only the wrist temperature and its changing rate (combined with gender and body mass information), an accuracy of 94.39% is estimated.

## 3. Building Simulation Models under the IoT Paradigm for the Study of Thermal Comfort

The literature about building energy models is not the object of this review, as its great extension would make this task almost impossible. Only papers that use building simulation as a tool to study the thermal comfort will be analysed and presented, always remaining under the IoT’s sphere of influence. The most common of the simulators have been seen to be EnergyPlus. Unless otherwise noted, the IoT function in these studies is represented by the sensing network layer used to collect data and to validate the models. One of the aims of this section is to investigate how the approach towards people is changing and developing, from a passive role to an active role, preparing the basis toward the people-centric approach of Crowdsensing.

Thermodynamic modelling is a highly popular procedure in building physics. There are several instruments to simulate buildings and their thermal environment. One of the most widely used is EnergyPlus [65,66]. EnergyPlus contains a specific section dedicated to the evaluation of occupants’ thermal comfort. It also allows the comparison among Fanger’s model (through PMV and PPD), adaptive model and other less-common thermal models. Several studies investigated the accuracy of EnergyPlus (EP) results in fields such as energy consumption [67,68], energy retrofitting [69,70], thermal resiliency [71,72] and, of course, thermal comfort [73,74]. Major contributions are summarised in Table 3.

### 3.1. Evaluation of Thermal Comfort under Modifications of the Physical Characteristics of the Building

The first group of EP studies concerns improvements in thermal comfort through changes in the building envelope. People are considered as passive receptors of the improved indoor thermal condition.

Rincón et al. [73] used EP to study the thermal comfort in earthen dwellings in sub-Saharan Africa (Burkina Faso) and underlined that the thermal stability inside this kind of building assured better thermal comfort perception of the users. They based their method on the evaluation of the so-called Discomfort Degree-days, that gives a more realistic way of estimating thermal comfort in a building. The *earthbag* building proposed in the paper shows a 76% improvement in thermal comfort condition compared to traditional Burkinabe dwelling (217 vs. 923 discomfort degree-days).

Ashrafian et al. [68] studied the potentialities of improvements in children’s thermal comfort and productivity at school, through changes in terms of the glazing ratio of the building. Their proposal is based on the increase of natural lighting and on a reduction in terms of HVAC energy consumption. According to their results, fewer than 20% of children would be thermally uncomfortable in almost all scenarios. Although, the study underlines that children are more sensitive to higher temperature than adults. Actually, none of the scenarios seem to be acceptable and PMV hardly reaches −1 (slightly cold), so other improvements should be included.

Oliveira and Labaki [70] studied the effect of thermal discomfort in the university environment. To reduce the physiological and psychological undesired effects of extreme temperature, namely somnolence, sweating and apathy, they proposed a retrofit of a university campus based on the insertion of a solar chimney, simulated in EP for the experimentation. According to the adaptive method, uncomfortable hours would diminish from 97% to 75% with the retrofit improvement.

Following EP configurations, people are also important heat emitters classified as internal loads. Esteves et al. [74] based their research on this concept. They simulated PMV and PPD of a cinema room and validated their model with experimental data collection, concluding that the difference in terms of dissatisfied people is smaller than 2%. They pointed out that the occupancy rate is a crucial factor when calculating thermal comfort; in fact, in the second session, the energy accumulated from people is enough to heat the cinema room, improving the PMV from slightly cold to neutral. The paper has just a monitoring aim, but it is included in this section because of its focus on people’s contribution to the thermal environment.

Kwok et al. [75] studied the potentialities of crossed-ventilation in hot-humid climates, analysing thermal comfort dependency on envelope composition. In hot-humid climate, the PMV model fails to describe comfort temperature, because it does not consider the effect of people’s acclimatization. For this reason, Fanger and Toftum [12] elaborated another model which included an expectancy factor, *e*. For instance, for the case study of a middle-floor flat in Hong-Kong, the authors fixed *e* = 0.7. According to EP simulation, natural ventilation does not solve the cooling problem in the warmest climates, since none of the examined models assured a thermally comfortable environment for more than 40% of the time. The simulation gave another interesting result: flats with better insulation guarantee a reduction of extreme indoor condition, but also imply a longer duration of time in which users feel uncomfortable.

### 3.2. Evaluation of Thermal Comfort Under Modifications of Users’ Behaviour

In this section, studies that proposed changing people’s habits are presented. The users’ role switches from passive receivers of environmental changes to the determining factor in the process of improving their own thermal comfort. This configuration involves changing users’ conscious control over their surroundings. For example, it is very common in experiments to let workers have an influence on the thermostat set-point in an office building, giving them control over the mechanically ventilated system of the workplace [76,77]. Likewise, passive strategies that depend on the manual and active intervention of users are also diffused [78,79]. In most cases, the change is made possible by alerts that sensors connected to the internet send to occupants’ devices, like smartphones or tablets.

Yun et al. [76] used their EP model to demonstrate that when occupants have a higher level of perceived control, their thermal acceptability tends to increase. They modelled seven university buildings in South Korea under mixed-mode condition. For a whole summer period, 77 users took part in the experimentation, answering a questionnaire about thermal perception. In some buildings, people were allowed to change the room temperature through a normal thermostat. After investigating the effects of operative temperature on the subjective evaluation of the users, the authors created similar EP models in which the only difference was the level of perceived control of the users, to evaluate the data coming from the IoT devices. In the model, the thermostat set-point is chosen according to the monitoring results and the answers to the questionnaires. The comfort temperatures for people with a higher level of perceived control was higher, allowing a 9% energy-saving and greater self-reported productivity of 6.3%. This is an example of how psychological factors can influence the thermal perception of people. Giving control, hence responsibility, to the occupants helps to raise awareness about their own adaptability, allowing them to be conscious of their real and personal thermal comfort zone.

Another study about personal control is the one by Kinnane et al. [80], conducted in dementia-friendly dwellings. The authors proposed a shift from active to passive design, that allowed greater personal control to the users. They used a questionnaire to investigate people’s opinion (aged between 79 and 82) and concluded that the ingress of cold air creates an important variation in thermal conditions and extreme discomfort in aged people. The cold air ingress was simulated using EP. This study points out one of the reasons for physiological differences among individuals—age. Age, gender, body structure and so forth are factors that determine the individual perception of the thermal environment. Hence, investigating thermal comfort in aged people and giving them control over their environment is a way to assure and increase the achievement of a comfort condition.

Pellegrino et al. [78] studied behavioural implications of users to achieve the thermal comfort in Indian residential buildings. Their method was based on the evaluation of the Comfort Indoor Degree Hours (CIDH), defined as the sum of the hourly differences between indoor air temperature and comfort temperature, and another index called Tropical Summer Index (TSI), that defines the tropical setting. Great importance is given to occupants’ behaviour, demonstrating that a diminution of 60% in CIDH can be achieved just with a conscious use of nocturnal ventilation. The authors also underline the importance of developing feasible and shareable solutions, because high-tech innovation could be far from people’s possibilities in some zones of the world. About behavioural implication, other considerations are interesting: air conditioning systems in areas like India are a social status, so people tend to reject the proposals based on passive strategies. Besides, nocturnal window-opening does not allow them to protect from external noises and artificial lighting. So, the authors had to face a paradox: although according to the obtained TSI index indoor condition was comfortable, occupants felt uncomfortable anyways and decided to install an AC. The last two studies brought out how psychological and physiological differences affect thermal comfort. In this case, another strong factor is found—the social one. Giving control to people is not enough without a phase of empowerment through information. People need the tools to understand why a solution is better than another and what benefits they can obtain by changing their behaviour.

About passive strategies, Thravalou et al. [79] confirmed that occupant behaviour has a large impact on the whole performance of the building. They studied vernacular architecture in Cyprus, simulating two types of natural airflow and five different natural ventilation strategies in summer. Using the adaptive thermal comfort model, they concluded that the best solution for that context was cross-ventilation during the night, with even greater results than in an all-day ventilation case. They conducted a survey to confirm the results about thermal comfort and 80% of occupants considered the obtained indoor thermal condition satisfactory. This paper demonstrates to the reader that overcoming the problem of lack of information is feasible. The authors explained to occupants which was the better solution for their specific thermal environment and how to put it into practice. Consequently, they managed to control their surroundings and to achieve a thermally comfortable condition.

The final study presented represents a summary of the considerations made about personal control and, at the same time, a link to the next chapter in terms of sensing occupants’ preferences through IoT. Jeanblanc et al. [77] analysed how occupants use energy in buildings in the United States, sustaining that reaching and maintaining the thermal comfort requires a dynamic interaction between the occupants and the building. The interaction is obtained through IoT devices, that provided both real-time data collection and feedback to users regarding their behaviour. They presented examples of natural ventilation potentialities, studying a research lab through the components that are operable by users (windows, louvers, sunspace). Operations considered three factors, namely facilitate passive ventilation, respond to weather conditions and mitigate solar radiation. As a consequence, they proposed five experiments to reduce the AC system use. The experiments showed an 80% compliancy of the users’ thermal comfort (adaptive model) in almost all cases, with 21% to 100% in the 90–100% compliant zone. Only one case presented as 8% outside the comfort zone, probably due to high humidity of a rainy day. The satisfactory results demonstrate the importance of a connection between humans and the thermal environment, only feasible through IoT devices. Monitoring plays a very important role, but the step forward is represented by feedback. Thanks to feedback, users are able to experiment with how their behavioural changes imply great differences in their surroundings.

### 3.3. Combined Platforms

In previous sections, IoT was limited to a sensor networking, since the case studies proposed were based on changes in envelopes or people’s habits and did not require any other technology. In this section, the examined case studies are based on a more complex IoT architecture, which includes other components such as smart thermostats. This implies the creation of integrated platforms that combine EP simulation and data processing with mathematical software.

Zhao et al. [81] dealt with the thermal comfort matter through the conception of a predictive control model based on the dynamic simulation and the data analysis with Matlab. They firstly created a control model based only on EP predictions; then, they conducted a users’ feedback experiment on 15 volunteers and used the results to develop an occupant-oriented model. The ultimate step forward is the implementation of a real-time thermal comfort feedback system, which can enable individual comfort preference and energy impact. The aim of the research is to save energy while maintaining thermal comfort, so that only results in terms of HVAC consumptions are shown.

Likewise, Kalvelage et al. [82] proposed a coupled use of an agent-based model (ABM) and building simulation with EP to estimate the effect of the task-based users’ behaviour on building energy demand. Nevertheless, they focus on overall comfort in their questionnaire, so results about thermal comfort cannot be extrapolated and will not be included in Table 3.

About control, research by Nouvel and Alessi [83] is also noteworthy, which presented a co-simulation through EP and Simulink to create an HVAC control algorithm based on a User Interface (UI). It contains six command buttons that correspond to the thermal scale used by Fanger, without the neutral sensation. Results showed an accurate correspondence between the actual thermal sensation of the users and the thermal sensation calculated by the algorithm. In both cooling and heating simulations, thermal comfort is improved, and energy waste is reduced (up to 57% in summer, and up to 22% in winter).

Another important contribution is the one of Escandón et al. [84] who studied a social housing stock in southern Spain to explore the possibility of evaluating the thermal behaviour of every user, without increasing computational time. They elaborated a “model of the model” to reduce computational times. To reach this scope, they theorized an Artificial Neural Network (ANN), generated under Matlab using EP-simulated data, that was composed of an input layer, a hidden layer and an output layer that provides the annual percentage of discomfort hours. According to their results, over the whole year, discomfort hours are quite acceptable, between 20% and 50%, but in winter, they reached an impressive range of 95–100% of discomfort hours. With the model created, they could extrapolate which are the critical issues in the stock, namely the geometry itself of the flat and the natural ventilation rate.

Ramallo-González et al. [85] proposed an integrated IoT infrastructure based on the FIWARE platform (an open-source platform for smart cities) to obtain a change in occupants’ behaviour to maintain thermal comfort while improving the management of energy consumption. The IoT energy platform, called IoTEP [86], was the first holistic solution for the management of IoT energy data. The infrastructure communicates with the user through a mobile app, to provide feedback and to induce a change in habits. The building understudy was modelled through the mathematical software Octave and simulated with EnergyPlus. They demonstrated that an effective change on all behaviours led to a 41% saving in the heating demand and an 8.3% saving in cooling demand.

Pombeiro et al. [87] proposed a Demand Response (DR) algorithm to explore the thermal comfort of users while aiming to save energy. They created two optimization models to reduce energy consumption, that are then compared in terms of users’ thermal comfort. A Simplified Thermal Model (STM) is created considering internal loads, heat exchange of the envelope, ventilation and radiation from surfaces. After the validation with the EP model, they included comfort parameters such as the PMV, the PPD and the accumulated energy by the human body (S). Results show a comparison among several tests that present the variation of the importance of users’ comfort on cost, searching for the best compromise.

Also linked to the DR algorithm is a study of Cetin et al. [88], which explored a good strategy to maintain thermal comfort while reducing energy consumption, namely the on/off controller of the HVAC thermostat. The gap in the on/off cycling resulted to be an important lack in EP simulation. To implement that, the authors proposed an on/off cycling Energy Management System (EMS). They tested the system with the model of a typical residential building in Sacramento, California, achieving a reduction in the Normalised Mean Bias Error (NMBE) from 27.1% to 8.3%. Although the method’s aim is improving the simulation of thermal comfort and energy-saving, the case study is limited to the implementation of the algorithm and leaves the quantitative analysis of those parameters to further studies, which is why it will not be included in Table 3.

### 3.4. Response of Thermal Comfort to Future Overheating

Although studying future overheating is not one of this review’s purposes, measures used today to improve the users’ thermal comfort will not necessarily be enough in future scenarios, in which extreme temperature events will be higher and more common. Indeed, researchers manifested a growing interest in investigating this topic through EnergyPlus. The studies in this section will not be included in Table 3, as they use criteria not comparable with the others. The trend for future climates is based on the utilization of synthetic weather data that morph the probable climate scenarios in 2020, 2050 and 2080. Alternatively, it is possible to consider a series of the warmest years or parameterise the overheating metric.

Brotas and Nicol [71] simulated a mid-stories flat in London and predict the overheating using three criteria defined by the CIBSE TM52 standard [89], namely maximum hours where the operative temperature can exceed the maximum temperature, the daily limit for acceptability of overheating depending on its severity and absolute maximum acceptable for a room. Then, based on the results of the case study, they extended simulation to other European cities to show how overheating will affect them differently. Fosas de Pando et al. [72] presented a parametric building simulation with a case study of a well-insulated dwelling. They compared different overheating trends, concluding that the TM52 ones are the most appropriate. The parameters they considered were different weather files, thermal mass, glazing ratios, occupancy profiles, shading strategies, purge ventilation strategies and orientation. Results showed the differences between different building fabric performances, to explore how the fabric choice changes the resiliency over the years.

Eames and Shorthouse [90] simulated a single-story building operating in free-running mode to compare metrics based on air temperature and radiant temperature. To determine the design extreme weather years, they utilized the return period method, while the parameter chosen to analyse the thermal comfort is the PMVH, i.e., the number of hours where the PMV is greater than 0.5. They considered the expansion of their method, used to calculate the design summer years for London, to the rest of the UK. Nevertheless, changing locations involves changes in weather patterns, which is why the risk of overheating should be dependent on location and metric.

## 4. Mobile Crowdsensing

Apart from IoT devices with sensing capabilities that can be environmental (air temperature, relative humidity and so forth) and physiological (skin temperature, heart rate, etc.), it is possible to find another way of acquiring data. This is Crowdsensing. Currently, almost everybody has a personal smartphone, and this can be a rich resource for the acquisition of data about thermal comfort and others. The first and most cited overview about mobile Crowdsensing (1638 citations in 2020, according to Google Scholar; 1100 citations according to Scopus) is “Mobile Crowdsensing: Current state and Future Challenges” by Ganti et al. [36]. In this paper, the term Mobile Crowdsensing (MCS) is coined to refer to the variable reading that a mobile device can offer: inertial situation, compass data, Global Positioning System (GPS), microphone readings, camera views, proximity or light. MCS applications are divided into three categories according to what is monitored: environmental, infrastructural and social. For example, in social applications, people share sensed information, e.g., their activity data, diet data, etc.

The unique characteristics of MCS applications according to Ganti et al., compared to traditional mote-class sensors, are:Today’s mobile devices have better resources in terms of communication, computing and storage than mote-class.People already use these devices whatever they do.Human intelligence can help to collect higher quality and more complex data than machine software.The incentive mechanisms behind this kind of application are normally strong.

Also, Ganti explains that Crowdsensing can derive from both participatory sensing, i.e., involving people’s active contribution and collecting their opinion through questionnaires, and opportunistic sensing. The main idea behind the application of opportunistic sensing is to use people’s phone as a sensor. To achieve this purpose, the key is the use of the actual sensors embedded in the smartphone. These sensors ideally will be upgraded over the next years including, for example, barometer, temperature and humidity sensors [91], whose potential has been shown by Choudhury et al. [92] through developing a Mobile Sensing Platform (MSP). As explained by Ali et al. [93], thanks to the expansion of sensor technology in the last decade, external devices can be connected to the mobile phone to track information about environmental parameters and body heat data from the user as well, multiplying the sensing capabilities of phones.

In the last decade, Crowdsensing has started to gain the interest of the scientific community. The discussion is open and has the potential to keep growing, but the investigation has already reached some interesting results. Leaving aside technological shortcomings (such as phone battery consumption and so on) and privacy concerns, this section will focus on people-related trends. The authors will briefly present some examples to explain how Crowdsensing can be applied to whichever field of study, from environment to infrastructure passing by society.

In a paper of Chon et al. [94], a framework called *CrowdSense@Place* (CSP) is presented. This framework is composed of a smartphone application and an off-line server to process the collected data. Data from locations and trajectories of users are used to link place visits with place categories (i.e., stores, restaurants and so forth). Sensors embedded in the mobile phone, namely microphone and camera, can catch certain “hints”, including words spoken by users, text written and objects surrounding the users. In this way, the CSP can automatically determine the type of place (e.g., gym), without the intervention of the user—this is an example of the aforementioned opportunistic crowdsensing. The experiment showed that the app was unable to categorize places with the accuracy expected (69% accuracy was obtained). Besides, privacy was considered insufficient to guarantee the utilization of the app by the general public.

Sun et al. [95] proposed an application of mobile Crowdsensing for tourism in the city of Trento, Italy. The data flow is collected both through smartphones and from smart wearable bracelets. This was included within a research project with the objective of promoting a sustainable cultural heritage in the city. Besides, the authors used the concept of Smart and Connected Community (SCC) to indicate an evolution of smart cities, extending its application also to urban science. Piao and Aihara [96] used motion sensors’ data to apply Crowdsensing to infrastructure. They aimed to monitor road surfaces, especially in snowy areas, through sensors embedded in smartphones, such as accelerometers and eventually cameras. Results showed that road surface conditions can be detected with a 90% accuracy. Similarly, El-Wakeel et al. [97] investigated road anomalies (cracks, potholes, manholes) to apply Crowdsensing to the route-planning field, using GPS both from vehicles and smartphones. When it comes to select a route, the most widely used planner tends to choose just based on time. This paper proposed an alternative method which bases decisions on road quality, analysed through Crowdsensing. The system, using the inertial sensors and GPS receivers available in all modern smartphones, was considered successful. Also concerning infrastructure is a study of Matarazzo et al. [98], that explored the problem of bridge vibrations using smartphones in moving vehicles. They aimed to combine several datasets to detect bridge modal frequencies of the bridge under study. The paper concluded that the smartphone was an adequate tool, in particular when heterogeneous datasets were aggregated to improve precision. This application can have an enormous impact on bridges maintenance, overcoming the traditional visual inspections method. On the other hand, interaction effects with the vehicle are not considered and the simple accelerometer included in the smartphone (without the aggregation) tends to measure imperfectly. Baljak et al. [99] applied crowdsensing to public transportation, collecting data from users’ smartphones through an app to study the vibration in a public vehicle in Belgrade. To improve the precision of the geolocation within the app, an implementation through Google Location Application Programming Interface (API), provided by Google, is proposed. Using accelerometers present in many modern smartphones resulted to be a good instrument to explore comfort in public transport.

With respect to indoor comfort, it is noteworthy to highlight the study by Angelopoulos et al. [100], in which Crowdsensing principles are applied to indoor lighting. They developed a testbed computer architecture to integrate crowd-sourced data resources, namely those coming from smartphones and tablets, to Smart Buildings. They aimed both to improve the sensing capabilities of the Smart Building with the embedded sensors of the aforementioned devices and to incentivise the direct interaction with people, asking for feedback on their experienced comfort.

Considering the studies proposed in this section, one can see how this new paradigm is catching on. It has great potentialities and it can be a powerful tool to give importance to people’s preferences. The authors firmly believe that Crowdsensing can also involve the field of thermal comfort, allowing users to be responsible for changes in their own thermal environment [101]. There are some first attempts to create platforms that unify the thermal comfort model and the crowd simulation, that will be discussed in Section 4.2.

### 4.1. Making Mobile Crowdsensing Work

If Crowdsensing is to be a tool for the study of thermal comfort, it is critical to have a good understanding of the incentives that are necessary to make people participate with this tool. Researchers in the last years have tended to focus on this incentive matter, developing several models. Yang et al. [102] proposed two models for defining incentives: the so-called crowdsourcer-centric model and the user-centric model. In fact, they theorised that users would not be interested in Crowdsensing without a reward because of their resource consumption (mainly battery) and their privacy concerns. In the crowdsourcer-centric model, the crowdsourcer chooses a fixed reward for participating users, while in the user-centric model, users place a minimum bid and then the crowdsourcer selects the subset of users to involve. Among other results, an important consideration concerns the optimal reward: they observed that it increases with the number of participants, but it gradually becomes steady as the number of users becomes larger. Nava Auza et al. [103] presented a framework that includes a model to represent both people’s behaviour and incentive method: different values of incentives are assigned based on the local information of users. Zhang et al. [104] proposed a framework called CrowdRecruiter to select participants through a piggyback task model, to obtain a minimization of incentive payments due to the reduced number of participants. A complete survey on the topic is provided by Zhang et al. [105], that defined three types of incentives in mobile Crowdsensing systems: entertainment, service and money. Services are intended as traffic monitoring, air pollution monitoring or noise monitoring, i.e., public service systems. Engaging based on entertainment consists of turning some sensing task into games: for instance, Morganti et al. [106] developed a game to increase pro-environmental behaviours.

Another way of convincing people to use a mobile Crowdsensing tool is through an engagement plan, as several social dynamics influence the success of a mobile app. Contrary to incentives with a reward, engagement plans let the users perceive that the app’s main scope is useful for them. Cottafava et al. [107] organised a communication campaign based on social media and official advertisement inside the building. Even so, they noticed a lack of response, so they concluded and suggested for future works to plan a communication strategy that involves the whole institution from the beginning. Sanguinetti et al. [108] proposed an engagement plan based on colloquial words, to be appealing to young users (for instance, “chilly” instead of “cool”). Furthermore, they featured a mascot (a cow) with a fun visual design, to make users empathise and connect with it emotionally. Finally, they carefully planned the promotion of the app in the campus, through events, giveaways, flyers and chalk drawings in the classroom. In this way, they received 900 sets of feedback just within the first month. However, participants expressed frustration when discomfort persisted, as shown by feedback such as: “Don’t ask for opinions on things you aren’t willing to fix”. According to both studies, interdisciplinary collaboration with professionals of design sociology or marketing is the key to maximise people’s engagement in the long term.

A communication and engagement plan is crucial for a good result in terms of diffusion and acceptance. As a lot of social dynamic influences the success of a mobile application, the main aim is to make the app desirable to the user. The creation of a database is not a good expedient, nor is scientific research. The users have to perceive that the app’s main scope is useful for them.

In a thermal comfort data-collection scenario, some of the output produced with the data collected can be shared with the users. For instance, sharing their personal neutral temperature and recording how it changes in different periods of the year could be interesting for them. As well as giving as an output the whole month visualisation, to record in which days the user suffered discomfort and assuring them that this amount is under a certain health percentage. This could be a good strategy to convince people to repeat the test frequently, in different locations and with different climate conditions.

### 4.2. Crowdsensing for Thermal Comfort

This new paradigm has great potential in its application but, up to now, thermal comfort seems to be underexploited, to the best of the authors’ knowledge. In fact, if one makes a Scopus search with the keywords “Crowdsensing + thermal + comfort”, one obtains zero results, while the keywords “Crowd + sensing + thermal + comfort” gives just one result [109] (accessed on Scopus, April 2020). Nevertheless, there are indeed studies moving toward this direction, which is why we grouped the first approaches in this section. The relevant studies are summarized in Table 4.

First of all, it is possible to define two different perspectives when approaching the topic. In fact, the word itself has been interpreted in the literature in two main ways. On the one hand, as Guo et al. [37] explained, the Crowdsensing paradigm is interpreted as a consequence of the participatory sensing, inasmuch both of them sense data from mobile devices to empower ordinary citizens. On the other hand, many researchers have a vision strictly linked to the meaning of the word “crowd”, i.e., a disorganised group of people that gather together. In this perspective, the Crowdsensing for thermal comfort is centred on how the movement of the crowd influences and is influenced by the thermal indoor conditions. This approach makes sense in places like commercial buildings, in which users are not stationary and their thermal environment cannot be described by a fixed sensor.

Such is the case in the study by Chen et al. [110], who used an agent map to analyse the responsive behaviours of people to the environmental stimuli. In particular, they analysed the heat thermal balance of people in a shopping mall, with the purpose of studying the movements of the crowd. The authors certainly have the merit of being among the first ones considering the impact of factors like clothing under the Crowdsensing paradigm. Differently, Chiguchi et al. [109] investigated the thermal effect of the crowd in indoor spaces, starting from the consideration that, in a real commercial building in Japan, measurements presented 2 °C difference with or without crowding. They developed a new model to estimate the PMV based on crowd density and distance between sensors and human bodies. For the crowd density estimation, only smartphones’ accelerometers and microphones have been used. Chiguchi et al. assumed the thermal resistance of clothing according to the relationship with outside temperature, while metabolic rate and air velocity are calculated considering people walking at normal speed. Indoor temperature resulted to be the more complex parameter, so the authors conducted a fluid dynamic simulation with CFD 2015 software, which has implemented a PMV derivation component. Then, the results were validated through a real-space experiment with 7 users, with a maximum error of 0.4 °C. This category will be not included in Table 4.

Digging into the other perspective, the vast majority of studies that sense thermal database on users’ feedback through their mobile devices is related to the concept of participatory thermal sensing. One of the first contributions is given by Erickson and Cerpa [35], that proposes to improve occupants’ comfort using humans as participatory sensors. They developed a mobile application using the same 7-point scale indicated in the ASHRAE standard. Users have to submit a room request to vote the thermal comfort in the specific room they are occupying and then they have to vote depending on the thermal sensation they are experimenting in the room. Votes more recent than 10 min are not counted to prevent bias. The study included feedback of 39 participants throughout 5 weeks, in heating mode. In the first week, feedback about actual conditions were collected: 0% of users voted that they were satisfied or neutral. Then, a script was used to generate a real-time control strategy of the air conditioning, based on the lessons learnt in the first week of monitoring. According to their results, the method achieved 80% of users being “satisfied” or “somewhat satisfied”, 13% “neutral” and 7% “somewhat dissatisfied”. The authors considered a 100% satisfaction, since 0% voted “dissatisfied”. Furthermore, they concluded that just by adjusting thermal comfort, a 10% energy savings over the baseline strategy was achieved. It is noteworthy to underline the importance of monitoring in Crowdsensing for thermal comfort. As the objective is to respect individual differences in thermal sensation, one of the highlights of this work is the dedication to understand how a specific group of people act and react to thermal changes. With this starting point, it is possible to create a solution that fits for that cluster, and that would be different in another context.

Lam and Wamg [111] proposed a smartphone app, called CarryEn, that is connected to the Building Management System (BMS) to improve both thermal satisfaction and energy savings. The app aims to find the optimiaed set-point according to both the thermal preferences voted by the users and data collected from the BMS (initial set-point temperature, room temperature and occupancy). Votes are expressed using the ASHRAE 7-point scale. The authors conducted two experiments, one in a commercial building and one in the campus of their university. During several weeks, the set-point was changed with an interval of 0.5 °C in each adjustment. According to users’ reactions to these changes, the authors elaborated an algorithm to find the optimised temperature of the HVAC system. Through the use of CarryEn, an improvement in thermal satisfaction of 28.2% was achieved and none of the participants experienced extreme discomfort. Besides, 13% of energy-saving was achieved. In this case, it is important to notice how the new paradigm of Crowdsensing has the potential to communicate with the building. For buildings that are already smart, as in the case of BMS, a smartphone app can easily become part of the loop, allowing the users to make the most of the potentiality of the entire system.

Sood et al. [112] created a human–building interaction framework called the SDE Learning Trail, under the new Net Zero Energy Building (NZEB) paradigm. SDE refers to the building of the experiment, the School of Design & Environment (SDE) of Singapore. The framework included 35 stations across the building, and each station presented a Quick Response (QR) code. The QR code allowed the users to both vote for their thermal, visual and aural preference, and learn with interactive information. In this way, occupants and visitors could leave their feedback while learning about sustainability features. Stations were placed in proximity of fixed sensors that collected environmental data. Feedback was expressed through a 3-point scale to limit subjectivity. Most of the 616 users resulted to be comfortable, and seldom preferred a change to a cooler condition. An interesting conclusion was the division of users in clusters of preferences that can be used to personalise special recommendations based on their past votes. Inserting visualisation components in the loop could really make the difference when it comes to sensitising people to the thermal matter. Thermal comfort and energy consumption are abstract concepts to the normal occupant, so translating them into a language that people can understand should be the key strategy to reach a people-centric approach.

Sanguinetti et al. [108] investigated the possibilities of upscaling a closed-loop participatory thermal sensing program, to involve a whole university campus. They aimed to inspire occupant’s participation, interpret the data collected and improve campus conditions. They developed a project called TerMOOstat, in a first phase through the university portal and, in a second phase, through a mobile app. Students and staff could express their thermal sensation on a 5-point scale with colloquial wording, and then compare their feedback with the others through a pie chart of total votes. Throughout 23 months, adequate thermal comfort was experienced by 11% to 40% of respondents, depending on the season. In another study [113], the same research group implemented a Wi-Fi system which connected thermostats with rooftop units (to collect outdoor air temperature), using Python scripts. They proposed three different thermostat modes: a constant set-point schedule, a strategy that reacted to users’ thermal comfort vote and an energy-efficiency strategy that considered both users’ vote and outdoor air temperature. As a result, they showed that with the third method, it was possible to save 20–30% of energy use without compromising comfort. This work marks a fundamental point. Smart thermostat and personal comfort are already a reality in some limited contexts, for example, in some smart homes. The effort that this research group made, discussing about scaling, will open a path to the continuous extension of the application field of Crowdsensing. In other applications, for instance, Crowdsensing applied to urban science, scaling from the single home to the district leads to a better management of the matter at all levels. Sanguinetti et al. [108] demonstrated that this would also be possible working with thermal comfort.

Cottafava et al. [107] conducted a multidisciplinary study, involving sociologists, physicists and computer scientists to create an IT infrastructure with the crowdsensing approach. The IT structure included a smartphone app, a Wireless Sensor Network (WSN), an online dashboard and some predictive algorithms. Their declared aim was to act both on users’ awareness and on the HVAC system, because of the constant interaction between people and the buildings. To do so, they proposed different scenarios for energy simulation: the standard one, the “unaware” one and the “aware” one. They obtained good results for all, increasing in thermal comfort and energy-saving, describing how users’ opinions would be weighted according to time spent in the room or metabolic activity. The authors understood the Crowdsensing perspective, that is not just a method to acquire information, but has to be a tool to make the information comprehensible to the users, making them aware of the impact of their actions on the energy system and on themselves, to achieve an active change in people’s behaviour.

Li et al. [114] proposed a complete architecture which acquires data from both human and environmental factors to develop a personalised HVAC control framework. The system collects environmental data through room sensors and human data, both physiological and behavioural, through wearable health monitoring devices. A phone application allows the user to have all these data displayed. A Python script constantly executes the decision algorithm to dynamically change the set-point of the HVAC system. The experimentation presents two case studies, one in an office building and one in three single-occupancy rooms (it is one of the few cases in which thermal comfort in residential dwellings is tested). They demonstrate an 80% accuracy of the method in predicting thermal preference both in natural ventilation and mechanical conditioning, and they conclude that their algorithm can reduce the uncomfortable reports by 53.7%. Li et al. did not mention the Crowdsensing paradigm, nor the participatory sensing, but the study has been included because it reflects all the characteristics of the new people-centric approach. The interesting thing about this study is that the research group took a step forward using a very common object: a commercial wristband. In the last years, the wristband achieved to engage people about matters that they ignored up to now, e.g., quality of sleep and health in their lifestyle. As a consequence, the wristband could also be used to raise people’s interest in their thermal environment.

Ramallo-González et al. [52] created a comprehensive IoT framework to improve communication between the building and the users. They applied the concept of ‘human sensors’, collecting occupants’ thermal preferences through a personal mobile app. Then, collected votes were used to provide a smart controlling of the HVAC’s thermostat. As a step forward, the app combines the Crowdsensing paradigm and Augmented Reality’s potentialities. In fact, to improve the comprehension of the HVAC system, users can visualise real-time consumptions from a particular device equipped with a QR code. The framework was virtually tested through simulations, showing that a control based on Crowdsensing is dependent on how the algorithms that aggregate the votes work. Hence, they concluded that the control system chosen influenced the amount of time needed to reach comfortable conditions for most users. The democratization of shared thermal spaces is a crucial matter if one considers how much time workers spend sitting at their desks. When asked why they do not express their thermal dissatisfaction, most people answer they do not want to bother their co-workers. They could easily reach a point in which everyone is uncomfortable, but nobody speaks. With a discrete vote system, like the one proposed by Ramallo-González et al., awareness about the topic is raised and users can freely share their preference through the mobile platform without bothering anyone.

Jazizadeh et al. [115] developed an Human Building Interaction for personalized Thermal Comfort (HBI-TC) framework for obtaining comfort preferences information directly from users through an intermediary—the smartphone app. When a user enters the app, the closest building to the GPS-based location is found. The authors used this app to identify the most influential ambient condition parameters: they conducted an experiment over two months in an office building equipped with sensors and BMS controllers. During this period, HVAC is operated according to users’ feedback, using a linear relationship between comfort preference indices and preferred requested temperatures. Moreover, simulations were run with synthetic data to consider the diversity of occupants’ characteristics. The thermostat temperature is controlled to be equidistant from preferred temperatures of all occupants in the same thermal zone. Besides, the authors conducted a think-aloud experiment to establish which could be the best approach to express thermal preferences [116]. Literature shows that the traditional ASHRAE scale is subject to users’ interpretation. Hence, a slider approach scale was introduced, allowing participants to express a thermal satisfaction vote that varies from −50 (cooler side) to 50 (warmer side). Changing the satisfaction scale is certainly an interesting way to approach the matter: several studies underlined the difficulty of the users to vote. This is one of the causes of inaccuracy in thermal comfort surveys. The slider approach gets close to people’s way of thinking, allowing them to better express themselves. The highlight of the work can be considered the use of synthetic data to underline the diversity of occupants: several studies addressed gender, age and body size differences as relevant when it comes to analysing the thermal preferences. Simulation and synthetic data are a good tool to explore these parameters.

## 5. Quantitative Analysis of Scientific Community’s Contribution

Considering the novelty of the topic at hand, the authors considered it interesting to perform a quantitative study of the publications that have been found in the literature. A trends analysis was conducted using the Repository Scopus to examine researchers’ interests in the topics of this review. Starting from the general to the specific, the first step of the study was searching for the keywords “Crowdsensing”, “Internet of Things objects” and “EnergyPlus”, the three categories of this study. Figure 3 shows how Crowdsensing is a growing phenomenon, that started from zero and grew steadily. Its magnitude is slightly hidden by the trends concerning IoT, that started its escalation some years before and embraced many more fields from the very earliest stages. Nevertheless, if one looks at the period 2010–2013 of the IoT curve, it is possible to notice a great similarity with the period 2015–2019 of the Crowdsensing curve, so that there are all the conditions for this new paradigm to grow in the future, almost as far as IoT.

Taking a step forward to the thermal comfort topic, Figure 4 shows the trend of the topic discussed in this review: “Crowdsensing + thermal comfort”, “Internet of Things + thermal comfort”, “EnergyPlus + thermal comfort”, “sensing + thermal comfort + indoor” and “sensing + thermal comfort + Internet of things”. The general trend in the last decade underlines a growth for all categories. Crowdsensing application to the thermal comfort area, as the authors discussed in former sections, is an almost inexistent field of study—only a few pioneer studies embraced this path, leaving a considerable growth margin for the next decade. The authors also investigated the queries “sensing + thermal comfort + indoor” and “sensing + thermal comfort + IoT” to include studies that are moving towards this direction, but that simply did not use the word Crowdsensing. Some of these documents have been discussed in the previous sections. Apart from the queries in the graphic, others such as “crowd sensing + thermal comfort”, “crowdsourcing + thermal comfort”, “IoT + thermal comfort” and “sensing + thermal comfort + IoT” presented documents that were already included in the previous research. Other queries have been discarded; for example, “EnergyPlus + thermal comfort” gave results that refer to zero energy buildings, “sensing + thermal comfort” contained papers about the use of sensors in general, likely “sensing + Internet of Things” and “sensing + IoT” and “crowd + thermal comfort” showed results about group distribution and sport facilities.

To analyse trends in more detail, a search grouped by year throughout the last decade has been conducted. Results are shown in Figure 5.

The main topics of the three macro-areas “IoT objects + thermal comfort”, “EnergyPlus + thermal comfort” and “Crowdsensing” (differentiated by colours in the Figure 5) have been investigated in Scopus year by year, from 2011 to 2020. When it comes to analysing research results, Scopus allows a division of documents by subject areas. The subject areas have been selected based on the number of documents and the closeness to the main topic of this paper (represented by a line of the same colour as the corresponding macro-area). The three subject areas selected were “Engineering”, “Energy + Environmental Science” and “Computer Science”. Then, among every subject area and every year, a quick reading of titles and abstracts of all the corresponding documents has been conducted and a main trend topic has been elected (labels in the Figure 5). Although some topics remained recurrent for more than one year, repetitions have been avoided to maintain the clarity of the picture. As a consequence, some years are not associated to a main topic. Contrarily, in the case of the “Crowdsensing” macro-area, the corresponding line for “Energy + Environmental Science” is intentionally left empty until 2015, when the first papers on these subjects started to appear.

The graph in Figure 5 shows how the discussion evolved in the last years in order to arrive at the trends analysed in this review today. It is clear that there is a general tendency to change the management of the thermal systems; among the main themes investigation is focusing on, real-time predictive controls based on user preference is quite recurrent in all the macro-areas. The attention to individual experience is particularly evident from 2015, in which trend topics like “Personal Preferences”, “Indoor positioning” and “Microclimates” are paving the way for a people-centric approach. Consequently, one can see a spread in human sensing (“Activity recognition” (i.e., predicting the movement of a person), “Low-cost devices”, “Participants recruitment” and so forth) and a consequent concern for people’s privacy. Starting from 2018, another step forward is evident, with the notion of “Human-driven edge computing”, that refers to a model which integrates the elements of humans, devices, internet and information: thanks to mobile Crowdsensing and participatory sensing, users can express their opinion and become a “sensor” themselves. This came together with the application of machine learning-based techniques, to insert the data sensed from people in the control loop. These trends were confirmed in 2019, in which documents focused on “Occupant-centric control”, “Personalised thermal comfort”, “Private data sensing”, “Optimisation-based control” and “Reinforced learning”. Besides, one can see that the pursuit of thermal comfort has been frequently matched with the environmental and ecological matter, especially with respect to the building simulation macro-area (“Environment and climate change”, “Household sustainability”, “Traffic and pollution”, “Impact on climate change”, etc.). This is understandable thinking about how most of the studies reviewed in this paper had the double aim of increasing thermal comfort and assuring a fair energy consumption. Many authors presented results about energy saving, sustaining that a correct management of thermal comfort often implies a reduction in energy waste. Besides, the clear path toward passive solutions and naturally-ventilated buildings (“Passive design”, “Naturally-ventilated buildings”, “Windows operations”) is strictly related to people’s behaviour and awareness, as discussed in Section 3.2. Hence, trend themes in 2020, namely “Predictive control”, “Sustainability”, “Real-time prediction”, “Adaptive sampling” and “Ecological solution”, are not surprising, being a direct consequence of the aforementioned dynamics. Interestingly, not only scientists seem to be aware of this change of point of view toward a user-centric approach, inasmuch as some trend topics in science, namely “Wearable devices sensing” and “Smart home”, are already widely in fashion in people’s actual everyday life (as seen in Section 1).

## 6. Discussion

The analysis in the previous sections shows a clear trend toward considering the one-size-fits-all method as inadequate. Considering trends in the very last years, some approaches can be considered outmoded; nowadays, it is almost obsolete to investigate users’ thermal comfort based just on the physical parameters of the environment, without collecting people’s sensations and preferences. It emerged that people’s individuality should have a central role in the discussion about thermal comfort.

Adjusting the environment values, i.e., changing the HVAC set-point temperature according to comfort zones defined by the standards, can improve global satisfaction but cannot help leaving someone aside. In the studies detailed in Section 2.1, where the solution proposed was a change in the environment condition, the thermal satisfaction is maintained around 80%, which can seem a good result, but it means that there remains a percentage of people who will be in a situation of discomfort most of the time. When it comes to project a new space according to standards, the indoor space is evaluated as adequate if some physical criteria are met. If air temperature and humidity range are in the correct values, people are supposed to be comfortable. No space is left for individual differences or preferences.

On the other hand, solutions based on expensive smart objects, like footwarmers or heating chairs, that of course represent an interesting response to the problem, do not approach the “principle of justice” at all. Theoretically, the aim of research should be the development of benefits of equal access to everyone, but solutions of this kind can certainly only be afforded by big enterprises or offices of a few countries, being less useful for the rest of the world. Contrarily, improvements based on simple solutions are more democratically usable, as testified by another huge trend easily recognisable among the studies considered: the propensity to passive and/or low-cost strategies.

In this context, building simulation emerged as a powerful tool to explore possible scenarios far in space (hot-humid climates, extreme climates) and in time (future previsions) while giving the possibility of modelling personal differences through metabolic activity and clothing level. The simulation gives to the researchers the great advantage of investigating without expensive hardware and, eventually, without big samples. What appears to be necessary from this part of the analysis is the validation of simulated thermal satisfaction results through questionnaires or through people’s votes.

Even so, incorrect use of simulation can lead to a lack of differentiation in thermal zones shared by several individuals. This is more evident in open workplaces, where every workstation could have a different microenvironment. The phenomenon can be caused by eventual proximity to doors and windows (airflow), glass walls (direct radiation), the air conditioning terminal and so forth. Homogenisation is due to the way software works. For instance, EnergyPlus tends to consider the thermal zones as homogeneous. It indeed allows to create a difference in mean radiant temperature for proximity to surfaces, but workplaces cannot be placed in their actual positions, since occupancy is equally distributed in the thermal zone. Moreover, the homogenisation involves the sensing process, since sensors are generally located in spots that are not representative of any specific conditions. Fixed sensors are positioned in a higher position compared to desks and, in most cases, just one sensor is available for an entire thermal zone. The phenomenon is even amplified in big indoor places like commercial centres, when the difference among microenvironments is enhanced by the heating emission of the crowd.

The authors believe that Crowdsensing can give a possible solution to these matters. Collecting people’s opinions through mobile phones would be an efficient way to give importance to individuality and to represent a unique scenario. The sensing process would consider the effective condition of users, avoiding time losses and errors due to the position of the sensors. Comparing simulated opinions with real ones could become a new way to validate building virtual models and to differentiate thermal zones in indoor open spaces.

In the examples of Crowdsensing applied to thermal comfort, it was evident how this method leads to an upside-down perspective compared to what has been the thermal comfort up to now, a perspective that is in line with a general trend of empowering users. Nowadays, we are in front of a series of drastic revolutions caused by apps and social media. For example, the popular wristbands led people to a change of habits in terms of health, quality of sleep, way of eating and so forth. Ten years ago, nobody cared about those parameters. Moreover, people are starting to show interest in environmental topics, i.e., climate change, plastic contamination and so on. Hence, the authors see a predisposition in users to also accept a change in their thermal habits. It could be a good period to push towards a mobile tool that emphasises the importance of thermal comfort. It would help people to have control over their surroundings on very different levels, both at home and at work.

Nevertheless, when analysing Crowdsensing literature, it emerged that the thermal comfort field represents a black spot among its applications. Searching deeper, it appeared to be a semantic matter: most of the studies that use mobile devices to ask for people’s opinion referred to Participatory Thermal Sensing instead of Crowdsensing. This was surprising, considering that the widely cited study of Ganti et al. [36] classified the participatory sensing as a branch of Crowdsensing: “We therefore coin the term mobile Crowdsensing (MCS) to refer to a broad range of community sensing paradigms”. On the same topic, in a study adequately entitled “From Participatory Sensing to Mobile Crowdsensing”, Guo et al. [37] stated: “The definition of participatory sensing emphasised explicit user participation when it was proposed. In recent years, with the development of mobile sensing and mobile Internet techniques, the scope of crowd problem-solving systems using mobile devices has been broadened. To this end, we extend the definition of participatory sensing from two aspects and term the new concept mobile Crowdsensing (MCS)”. A reason for this discrepancy could be the lack of embedded sensors to sense the environmental conditions; in the studies proposed, smartphones have been used as a fundamental intermediary to help people express their preferences, but their potential role as sensors is barely mentioned. A few cases reported the attempt of using microphones and accelerometers to sense this kind of data; for instance, the smartphone’s microphone was used by Nam et al. [117] to measure the airflow. Even so, calibration and validation are left to future works. Some concerns might be raised about the mean radiant temperature, but it is widely used in the scientific community as the simplification that sets the mean radiant temperature equal to the air temperature. In fact, it was demonstrated that the differences between Ta and Tr are negligible for most periods indoors [118]. These methods cannot be seen as the ideal ones in terms of accuracy, and this could be the reason why the Crowdsensing was not applied to the thermal comfort field up to now. Improving the embedded thermal sensors of the most used mobile devices (smartphones, tablets and wristbands) will be an innovative topic for future studies, allowing to take this final step towards the Crowdsensing for thermal comfort.

## 7. Conclusions and Future Work

This review aimed to analyse the literature about thermal comfort under the IoT paradigm produced in the last decade focusing in particular on the role played by users. The approaches to the topic were multiple: some studies prioritise the evaluation of thermal comfort, with monitoring purposing, some others based their investigation on simulations models and evaluated their accuracy, comparing with real scenarios, and finally, some researchers focused on designing strategies to improve the level of comfort. As discussed, changes could concern the whole thermal zone or the micro-environment surrounding the individual, and they could depend on the physical characteristics of the envelope as well as on people’s way of behaving indoors or the ventilation mode.

One of the most important conclusions that emerges from the analysis concerns the growing attention to the people’s control and responsibility. According to several studies, when the users are included in the loop and they are conscious of the energetic problem, they tend to adopt a pro-environmental attitude. Two main issues have been detected. Some researchers investigated people’s preferences in order to adjust the environmental conditions. These case studies show a greater interest in energy-saving than in the effective impact of their strategies on comfort. Some others gave blind control to the users over the thermal environment, but they did not monitor their management actions. The goal of achieving a pro-environmental behaviour is left apart. Finding a compromise would allow scientists to better understand people’s preferences and actions, and it would allow users to be empowered through a conscious control over their thermal surroundings.

The Crowdsensing paradigm does have the potentialities to face the user control and the awareness’ matters, considering, for example, that nowadays, 45.04% of people own a smartphone [119]. Other devices that can be useful to sense data from people, like wristbands or the nearable devices on the market, have affordable prices and are already in fashion in some developed countries. All the conditions let us think that this topic will be a trend in research during the next decade.

From the evaluation of different kinds of approaches, the studies that present a more complete and global perspective are those based on a platform that involves more than one method. In this way, the study of thermal parameters becomes a process in which data and results are constantly improved and optimised thanks to the comparison of different tools. Including people’s thermal vote in these platforms, as some researchers started to do, will be a fundamental step towards a user-centric view of the problem and towards obtaining more reliable results. The better way to guarantee a dynamic data flow resulted to be expressing the aforementioned votes through the use of specific mobile apps. For instance, the combined use of a Building Management System (IoT control system in buildings) and participatory sensing though a mobile app gave incredibly good results (100% satisfaction according to the authors), obtaining a dynamic set-point temperature that is equidistant to all user preferences.

Under this new system and through the implementation of the mentioned embedded sensors, people themselves would act as thermal sensors, modelling their thermal environment according to their sensations as individuals. Or, taking a step forward, they could become a tool to validate the modelling. Data flow collected through smartphones could be considered as a parameter to validate the simulated models in software like EnergyPlus. EnergyPlus models are commonly validated through comparison with the environmental data collected by sensors. It would be interesting to investigate whether or not thermal preferences could work as a guideline to validate models, in buildings not equipped with sensors.

Up to now, crowdsensing was not applied to the thermal comfort study because sensors included in modern smartphones do not have the required level of accuracy. Nevertheless, nowadays, new technologies improve rapidly, so the authors believe Crowdsensing will be the new horizon for the thermal comfort evolution under the IoT paradigm.

## Figures and Tables

**Figure 1 sensors-20-04647-f001:**
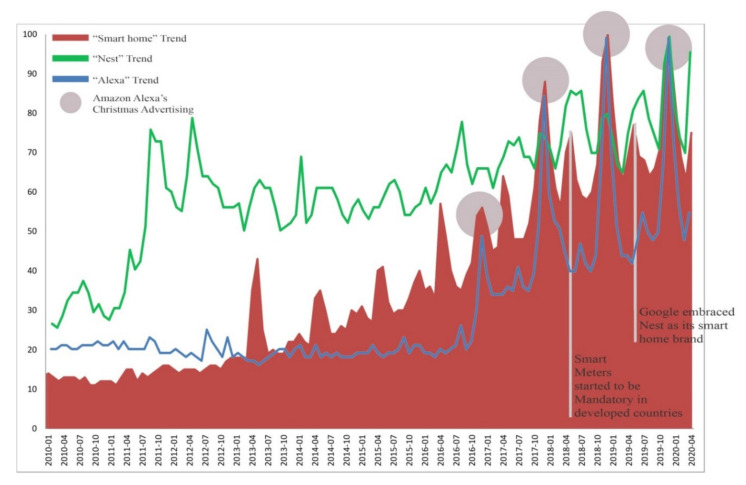
Normalized Google trends of the queries “Smart home”, “Alexa” and “Nest”.

**Figure 2 sensors-20-04647-f002:**
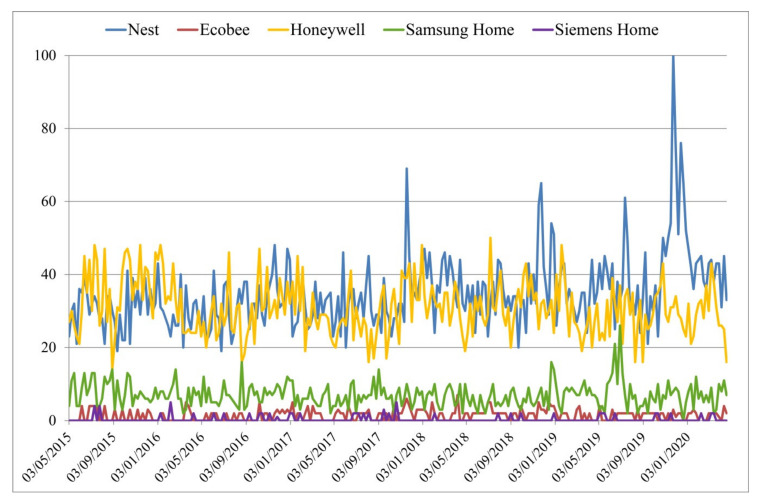
Trends in Google Shopping of the queries “Nest”, “Honeywell”, “Ecobee”, “Samsung home” and “Siemens home”. From Google Trends.

**Figure 3 sensors-20-04647-f003:**
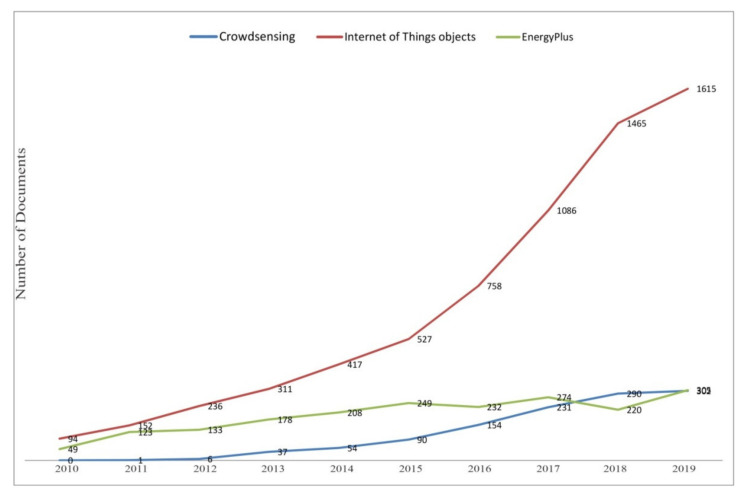
Trends of studies according to Scopus in the last decade. Queries: “Crowdsensing”, “Internet of things objects”, “EnergyPlus”.

**Figure 4 sensors-20-04647-f004:**
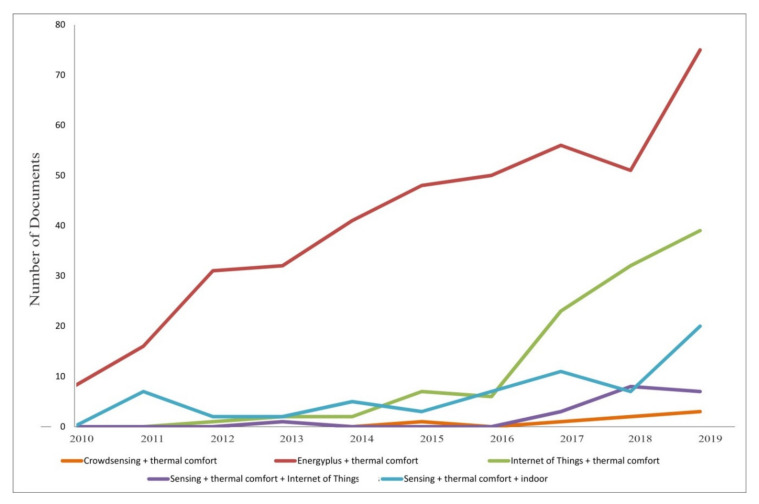
Trends of studies according to Scopus in the last decade. Queries: “Crowdsensing + thermal comfort”, “EnergyPlus + thermal comfort”, “Internet of things + thermal comfort”, “sensing + thermal comfort + Internet of Things” and “sensing + thermal comfort + indoor”.

**Figure 5 sensors-20-04647-f005:**
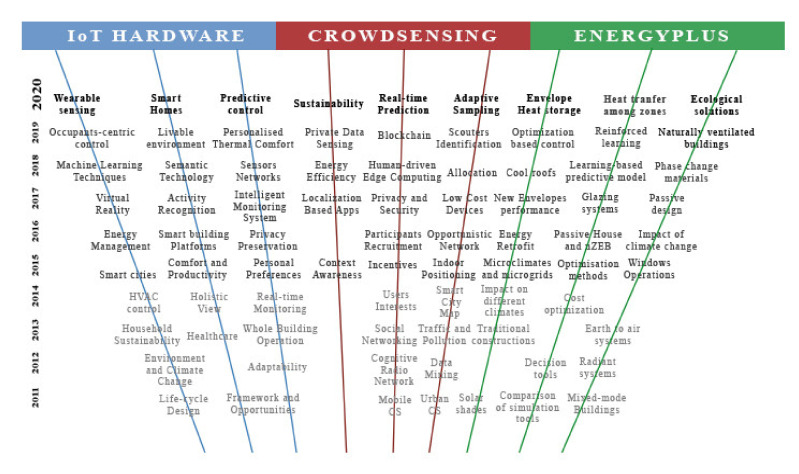
Tree of main topics based on thermal comfort in the last decade.

**Table 1 sensors-20-04647-t001:** Main contributions using Internet of Things (IoT) objects to enhance thermal comfort.

Source	Smart Objects	Thermal Comfort Model	Mathematical Model	Heating versus Cooling	Availability	Context	Sample	Geographic Location	Duration	Thermal Satisfaction	Energy Savings
Knecht et al. (2016) [54]	Personal devices, mostly garments	Adaptive	Inductive approach proposed by Braun and Clark (2006)	Both	Commercially available	Open-plan offices (university)	6 (heating) 8 (cooling)	London, UK	4 weeks (Mar) and 4 weeks (Jul)	Not quantified	Not quantified
Zhang et al. (2015) [55]	Footwarmers	ASHRAE Standard 55	Berkeley Simple Measurement and Actuation Profile	Heating	Fabricated by the authors	Office Workplace (university library)	16	Berkeley, California, USA	Small periods during half a year	80–100%	37–75% depending on the outdoor temperature
Feldmeier and Paradiso (2010) [49]	Smart HVAC, wearable wrist devices, sensors, control nodes	PMV Model (with minor modifications)	Hybridized control system	Cooling	Circuit boards fabricated by authors	Workspace (university)	10	Cambridge, Massachusetts, USA	Three months (May–Aug)	More than 80%	Up to 24% over the previous HVAC control system
Salamone et al. (2018) [57]	Wristband and nearable devices (sensors)	PMV model and adaptive model	Grasshopper, Python, Machine Learning	Heating	Commercially available	Office building	8	Milan, Italy	Small periods over 3 weeks (Nov)	Not quantified	Not quantified
Sung et al. (2019) [50]	Smart HVAC, sensors	PMV Model	Matlab simulation, Machine Learning	Cooling	Commercially available	Workspace	12 (simulated)	Taiwan	Not specified	83%	6–11.3% over the “comfort mode” (estimated)
Kim et al. (2018) [56]	Smart chairs	New PCS model	Machine Learning	Both	Fabricated by the authors	Office (university)	38	Berkeley, California, USA	7 months (Apr to Oct)	Not quantified	Not quantified

Acronyms: ASHRAE (American Society of Heating, Refrigerating and Air-Conditioning Engineers); PMV (Predicted Mean Vote); PCS (Personal Comfort System); HVAC (Heating, Ventilating and Air Conditioning).

**Table 2 sensors-20-04647-t002:** Main contributions using IoT objects to monitor thermal comfort.

Source	Thermal Comfort Model	Smart Objects	Mathematical Model	Context	Sample	Geographic Location	Heating/Cooling	Accuracy
Dai et al. (2017) [60]	ASHRAE Standard 55	Fibre Bragg grating-based sensors	SVM classifier, Machine Learning	Controlled Environmental Chamber + private office	11 (experiment 1) + 1 (experiment 2)	Berkeley, California, USA (exp 1) + Shanghai, China (exp 2)	Both	Over 80% (to 90% with 28 samples and three-inputs model)
Choi et al. (2017) [64]	ASHRAE –PMV survey designation	Exposed thermistor-type skin sensors	Excel, Minitab, stepwise regression	Experimental chamber at University of Southern California	18 (11 males and 7 females)	Los Angeles, California, USA	Both	95.87% with 3 body parts, 94.39% with one body area and the changing rate
Ghahramani et al. (2018) [59]	ASHRAE standard requirements	Infrared sensing system	Hidden Markov Model-based learning method	Shared office space university building	10	Los Angeles, California, USA	Both	82.8%

***Acronyms: SVM (Support Vector Machine).***

**Table 3 sensors-20-04647-t003:** Main contributions using the EnergyPlus (EP) simulation.

Source	Thermal Comfort Model	Heating/Cooling	Context	Geographic Location	Sample	Duration	Changing Proposal	Energy-Saving
Ashrafian et al. (2019) [68]	PMV model	Heating	Classrooms	Eskisehir, Turkey	12 classrooms	2 academic semesters	Preliminary design stage	8.5%
Escandón et al. (2019) [84]	Adaptive model	Both	Social housing	Seville, Spain	2 people	One year	Retrofit strategies	Not defined
Ramallo-González et al. (2019) [85]	Adaptive model	Both	University campus	Murcia, Spain	13 thermal zones	1 year (data collection)	Behavioural modification	41% Heating, 8.3% Cooling
Esteves et al. (2019) [74]	PMV model	Heating	Cinema Room (mechanically ventilated)	Penafiel, Portugal	6041 people (simulated)	2 months (Dec–Jan)	No changing proposed	Not quantifiable
Jeanblanc et al. (2016) [77]	Adaptive model	Cooling	Research lab	Iowa, USA	1 lab	3 months (Jun to Aug)	Natural ventilation	Up to 83%
Kinnane et al. (2016) [80]	PMV Model	Heating	Dementia-friendly dwellings	Dublin, Ireland	5 people (aged between 79 and 82)	Not specified	Personal control	Not quantifiable
Kwok et al. (2018) [75]	ePMV model	Cooling	High-rise residential building	Hong Kong, China	Simulated occupant density of 0.083 people/m^2^	Not specified	Natural ventilation	Not quantifiable
Nouvel and Alessi (2012)	PMV model	Both	Office building	Lyon, France	2 people (simulated)	1 week (summer) + 1 week (winter)	HVAC control architecture	57% (summer); 22% (winter)
Oliveira and Labaki (2016) [83]	Adaptive model	Cooling	University campus	Campinas, Brazil	1 office room	8 months (Dec to Aug)	Solar chimney	Not quantifiable
Pellegrino et al. (2016) [78]	“Model-free” approach	Cooling	Dwellings (naturally ventilated with a ceiling fans)	Kolkata, India	2 dwellings	1 month	Low-cost strategies and behavioural modifications	35% (flat 1), 76% (flat 2)
Rincón et al. (2019) [73]	Adaptive model	Both	Dwelling (naturally ventilated)	Burkina Faso, Africa	2 people (simulated)	3 weeks (one in Dec, one in Jun, one in Apr)	Passive strategies	Not quantifiable
Thravalou et al. (2016) [79]	Adaptive model	Cooling	Vernacular building	Nicosia, Cyprus	1 building	2 months (Jul–Aug)	Passive strategies	Not quantifiable
Yun et al. (2018) [76]	Adaptive model	Cooling	University building (mixed-mode condition)	Suwon, South Korea	77 people	3 months (Jul to Sep)	Perceived control	9%
Zhao et al. (2016) [81]	PMV model	Heating	Office building (mixed-mode)	Pittsburgh, Pennsylvania, USA	15 people	3 months (Oct to Dec)	Active HVAC control	Up to 61.20%

**Table 4 sensors-20-04647-t004:** Main contributions using Participatory thermal sensing.

Source	Thermal Comfort Model	Vote Scale	Heating versus Cooling	Context	Sample	Geographic Location	Duration	Changing Proposal	Energy Savings	Thermal Satisfaction
Lam and Wang (2013) [111]	PMV model	ASHRAE 7-point scale	Cooling	Commercial building and university	11 people (commercial building) + 12 (university)	Hong Kong, China	3 weeks (commercial building) + 4 weeks (university)	Optimised set-point	13%	28.2%
Cottafava et al. (2019) [107]	Adaptive model and PMV model	5-point scale designed by authors	Both	Classrooms and offices	Not specified	Turin, Italy	9 months	Optimised set-point	Up to 54%	From 1.7 to 2.7 on a 1–5 scale
Jazizadeh et al. (2014) [115]	PMV model	Thermal Preference (TP) scale, designed by authors (from −50 to + 50)	Cooling	Office Building	4 people and 7 simulated	Southern California, USA	2 months	Optimised set-point	Not quantified	Not quantified
Li et al. (2017) [114]	“Model-free” approach	Thermal sensation: 5-point scale, Thermal preference: 3-point scale	Both	Workplace, single-occupancy rooms	7 (office), 3 (rooms)	Wisconsin (office), Michigan (rooms), USA	3 weeks winter (office), 6 weeks summer (rooms)	Optimised set-point	Not quantified	Estimated reduction uncomfortable reports: 53.7%
Sanguinetti et al. (2017) [108]	PMV model	5-point scale designed by authors (ASHRAE inspired)	Both	University campus	4300 users	Davis, California, USA	23 months	Optimised set-point	20–30%	11–40%
Erickson and Cerpa (2012) [35]	PMV model	ASHRAE 7-point scale	Heating	LEED Gold-Certified Building	39 participants	Merced, California, USA	5 weeks	Optimised set-point	10.1%	80% “satisfied” or “somewhat satisfied”, 13% “neutral”
Sood et al. (2019) [112]	PMV model	3-point scale designed by authors	Cooling	Net Zero Energy Building	616 users	Singapore	3 months	None (monitoring aim)	Not quantified	Not quantified

**Acronyms: LEED** (Leadership in Energy & Environmental Design).

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
