# Peer review of "A Comprehensive Survey about Thermal Comfort under the IoT Paradigm: Is Crowdsensing the New Horizon?"

_sensors, 2020, doi:10.3390/s20164647_

Round 1
Reviewer 1 Report
I read the proposed article with an interest, I myself deal with thermal comfort measurements with the participation of users and the topic is close to my research. I believe that the topic of a practical solution to the issue of thermal comfort based on smart technologies and active participation of users is important and current.
The article is reasonably well written and reads fairly well. Authors showed a good and extensive review of the literature.
Nevertheless, in my general opinion, the authors do not argue, comment or discuss the cited research results (at place of citation), the literature is presented in a dry and schematic way, and I felt like I was reading several abstracts line by a line(e.g. secion 3.2 or 4.2) and after these long descriptions, the authors left the reader (me) alone without any puenta.
It would be much better in my opinion, however, if the authors had more insight into the results obtained by other researchers, presenting their own comments (guideline/narration to reader).
The introduction to the article should better relate to the content of the article and the authors' question presented in title.
In line 43-48 I suggest that chapters structure justification should not mix secion numbers with sub-chapters numers (1.1).
The Discussion section is missing from the entire article.
In the context of promoting thermal comfort Corowdsensing solutions, I believe that the authors should provide some information about the potential equipment / sensors to be used and the uncertainty of estimates/measures. As shown by the metrological analysis of the equipment used to measure thermal comfort the uncertainty of PPD measurement with a high class equipment is 3-4%. The uncertainty of determining the thermal comfort by the vote method is 8-15% depending on the size of the group. The complex uncertainty of thermal comfort is usually around 7-10%. So my question is how the uncertainty of measurements and voting may affect the applicability of the promoted Corowdsensing method? How accurate would be this method?
In addition, I ask what sensors should be used in a smart devices. The humidity and temperature alone are rather not sufficient to define the thermal comfort of people (the air flow and radiation/black globe temperature should also be taken into account) How can this technical problem may be solved?
Detailed comments:
- after line 285, the article did not have line numbering and page numbers, which made the review very difficult
- authors should replace "we" by "authors" throughout the article
-IoT appears in the abstract. The definition should already be there. Later, consistently I recommend using the abbreviation in the text without long "the internet of things"
-research [9] is already 20 years old. Is it still good?
-Line 62 "modern studies" - what do the authors understand by the word “modern” here?
-Line 76 - ASRAE 55 was revised in 2017 and EN 15251 was replaced in 2019. Please take this into account. Additionally, there should be no BS before EN 15251. Journal is not a British, so EN alone is enough.
-Line [33] - Is sociological research conducted in the 1950s [33] still relevant / relevant to subject?
-Figure 1 is made using google big data. Maybe it should be noted in text.
-line 240. PPD term needs to be defined. Then, the full description should not be used in the text in terms of PMV and PPD (e.g. Section 3)
-Figure 2 is in very low resolution
-241. "0 to 0.7"? or (-0.7 to 0.7)?
-Line 252. Value 2.6OC [53] will refer to "at rest" (including metabolic activity).
-Section 3.5. Conclusions should be in the section discussion or conclusions. Such a section makes no sense in my opinion.
-Google Maps (section 4) should have a reference in the literature.
-Section 4.1. requires more comment from the authors
-Section 5 (title) - “quantitative analysis” - of what?
-Conclusions- 7 lines from the top- the authors limit the factors to people's behavior and the quality of envelope parameters. How about air movement/ventilation impact?
- Conclusions- some sentences require stylistic improvement, e.g. “sensing people sensation”, “trend that trends”
-Conclusions- 80% of satisfied does not automatically cause 20% to be dissatisfied. It depends on the adopted rating scale.
-Conclusions - should better include the results of analyzes carried out by authors from previous chapters without introducing new elements.
- There are also numerous references to Section 0 (?) (like line 249)
Author Response
Thank you for your comments, they have been of great value, especially considering you deal with thermal comfort too.
Please see the attachment.

Reviewer 2 Report
The article entitled "A comprehensive survey about Thermal Comfort under the IoT paradigm - Is crowdsensing the new horizon?" proposes a review of the most influential scientific publications that mix the fields of Internet of Things and thermal comfort. However, some aspects should be improved for publication.
The article must appear as a revision in the title.
The summary is confusing and poorly structured. It is necessary to present the summary according to the structure expected for this, a brief introduction, the central objective of the research, the methods that were used to carry out the study and the results and conclusions reached.
The authors addressed a range of works and reviewed the state of the art of the last decade on the subject. However, it is necessary to have a chapter explaining in detail how this review was structured, which was the central question of the research, which were the research methods of the articles for the review, which keywords were used and on which platforms this research was.
After reviewing these details, I consider publishing the manuscript.
Author Response
Thank you for your suggestions, they have been of great value.
Please see the attachment.
